# Peptidoglycan recycling mediated by an ABC transporter in the plant pathogen *Agrobacterium tumefaciens*

Michael C. Gilmore [1] & Felipe Cava [1] ✉

During growth and division, the bacterial cell wall peptidoglycan (PG) is remodelled, resulting in the liberation of PG muropeptides which are typically reinternalized and recycled. Bacteria belonging to the Rhizobiales and Rhodobacterales orders of the Alphaproteobacteria lack the muropeptide transporter AmpG, despite having other key PG recycling enzymes. Here, we show that an alternative transporter, YejBEF-YepA, takes over this role in the Rhizobiales phytopathogen *Agrobacterium tumefaciens*. Muropeptide import by YejBEF-YepA governs expression of the β-lactamase AmpC in *A. tumefaciens*, contributing to β-lactam resistance. However, we show that the absence of YejBEF-YepA causes severe cell wall defects that go far beyond lowered AmpC activity. Thus, contrary to previously established Gram-negative models, PG recycling is vital for cell wall integrity in *A. tumefaciens*. YepA is widespread in the Rhizobiales and Rhodobacterales, suggesting that YejBEF-YepA-mediated PG recycling could represent an important but overlooked aspect of cell wall biology in these bacteria.

The bacterial cell wall is composed primarily of peptidoglycan (PG), a heteropolymer which consists of alternating *N*-acetylglucosamine (GlcNAc) and *N*-acetylmuramic acid (MurNAc) sugars crosslinked by short peptide chains[1]. In order to facilitate growth and division, as well as adaption to environmental stressors, the cell wall is constantly remodelled by an arsenal of lytic enzymes[2]: glucosaminidases and lysozyme cleave between the two sugars, endopeptidases cleave within the peptide chains, carboxypeptidases cleave the C-terminal amino acid from the peptide chains, amidases cleave the peptide chain from the MurNAc sugar and lytic transglycosylases (LTs) cleave between the sugars with simultaneous cyclisation of the terminal MurNAc to form 1,6-anhydroMurNAc. A consequence of this remodelling activity is the liberation of PG fragments, termed muropeptides, which can be released into the environment where they can have far-reaching consequences in signalling or interspecies interactions[3], or imported into the cell for reuse in PG synthesis, in a process termed PG recycling[4].

PG recycling has been well studied in some Gram-negative models. Typically, anhydromuropeptides liberated by LTs are imported by the Major Facilitator Superfamily (MFS) transporter AmpG[5],

before being broken into their constituent sugars and peptide by the amidase AmpD[6], β-hexosaminidase NagZ[7,8] and LD-carboxypeptidase LdcA[9] (Fig. 1A). Pathways exist for recycling both the anhydroMurNAc and GlcNAc sugars[4,10–12], while the peptide is directly reincorporated into PG synthesis by ligation with UDP-MurNAc by the ligase Mpl[13] ultimately resulting in the production of the primary soluble PG precursor, UDP-muramylpentapeptide (UDP-M5). Around 50% of the sacculus is turned over in *Escherichia coli* per generation[14], but only around 8% is released to the medium, due to recycling[15]. Despite the extent of PG turnover, the abundance of proteins produced solely to enable PG recycling and its relatively wide conservation, its importance is unclear, since its interruption does not affect bacterial growth under laboratory conditions[4]. The exception to this is a growth defect caused by the deletion of the LD-carboxypeptidase LdcA, which leads to a reduction in cell wall crosslinking due to the inability of tetrapeptide precursors to be used as a donor in the PG transpeptidation reaction[9,16].

We noticed that Alphaproteobacteria belonging to the orders Rhizobiales and Rhodobacterales lack AmpG orthologs, despite

[1]Laboratory for Molecular Infection Medicine Sweden (MIMS), Umeå Center for Microbial Research (UCMR), Department of Molecular Biology, Umeå University, 90187 Umeå, Sweden. ✉e-mail: felipe.cava@umu.se

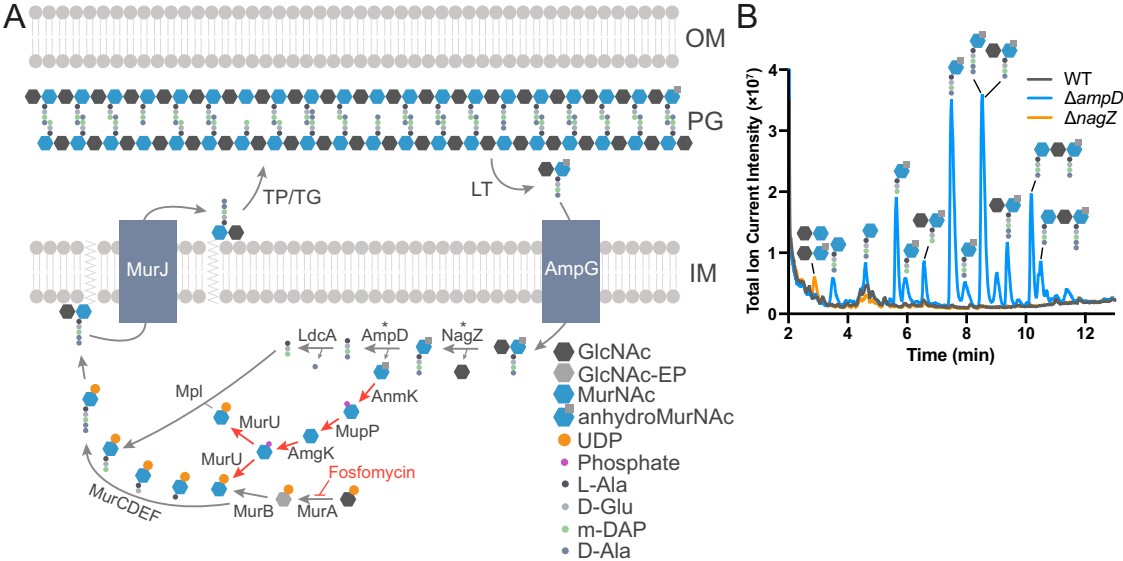

**Fig. 1 | *A. tumefaciens* recycles muropeptides. A** Schematic diagram of PG recycling pathway in *E. coli*. *Pseudomonas* MurNAc recycling pathway depicted with red arrows. EP enolpyruvate, TP transpeptidase, TG transglycosylase, LT lytic transglycosylase, OM outer membrane, PG peptidoglycan, IM inner membrane. * AmpD and NagZ activities are promiscuous: AmpD can act on muropeptides with or without GlcNAc[6], and NagZ can act on disaccharides with or without peptide chain[7,8]. **B** Total Ion Current (TIC) chromatogram showing soluble PG fragments that accumulate in the cytoplasm of *A. tumefaciens* WT, Δ*ampD* and Δ*nagZ* strains detected using LC-MS.

encoding orthologs of other PG recycling enzymes such as AmpD and NagZ in their genomes. Using the plant pathogen *Agrobacterium tumefaciens* as model, we show that the broad-specificity peptide ABC transporter YejBEF takes over the role of AmpG as muropeptide transporter in these bacteria, utilising an alternative PG-specific substrate-binding protein (SBP) YepA which is unique to the Alphaproteobacteria. PG recycling mediated by YejBEF-YepA governs expression of the β-lactamase AmpC through a canonical AmpR repression system, controlling β-lactam resistance in *A. tumefaciens* in place of AmpG. Importantly, we show that contrary to previously established Gram-negative models, PG recycling is vital for cell wall integrity in this bacterium since deleting the transporter results in significantly altered cell wall density and composition, leading to growth defects on low-osmolarity medium and hypersensitivity to β-lactam antibiotics in a manner partly independent of AmpC.

## Results

### A. tumefaciens recycles PG

We first set out to determine if *A. tumefaciens* recycles PG, since it lacks an ortholog of the PG recycling transporter AmpG. Since deletion of the anhydromuramyl-amidase AmpD results in the accumulation of anhydromuramyl tripeptides in the cytoplasm of *E. coli*[6], we deleted the *A. tumefaciens* orthologs of both *ampD* (Atu2113) and *nagZ* (Atu1709) and used ultra-performance liquid chromatography coupled to quadrupole-time of flight mass spectrometry (UPLC-MS) to check for the accumulation of the substrates of these enzymes in the cytoplasm (Fig. 1B). We observed the accumulation of anhydroMurNAc- and MurNAc-linked muropeptide species accumulating in the cytoplasm of Δ*ampD*, while the GlcNAc-anhydroMurNAc and GlcNAc-MurNAc disaccharides accumulated in the cytoplasm of Δ*nagZ*, confirming these enzymes play the same respective roles in PG recycling in *A. tumefaciens* as in other Gram-negative bacteria. Interestingly, we observed tri-, tetra- and pentapeptide-containing muropeptides accumulating in Δ*ampD*. In *E. coli*, typically only tripeptide muropeptides are observed to accumulate in an *ampD* mutant due to the activity of the LD-carboxypeptidase LdcA[6], suggesting that *A. tumefaciens* lacks this activity in its cytoplasm. To confirm this, we ectopically expressed LdcA from *E. coli* in *A. tumefaciens* on a plasmid

and observed a corresponding shift in accumulation from tetrapeptide to tripeptide muropeptides (Fig S1). In addition, trisaccharide muropeptides also accumulated in Δ*ampD*, suggesting that *A. tumefaciens* can transport dimers into its cytoplasm for recycling. The ion masses of these were consistent with either DD-crosslinked dimers or linear (uncrosslinked) chains. The possibility of these being LD-crosslinked was excluded because LD-crosslinked dimers include a tripeptide stem as a result of the LD-transpeptidation reaction[17]. LdcA expression allowed us to establish their identity as linear dimers as it resulted in di-tripeptides (i.e. M3-M3, Fig. S1), instead of the M3 and M4-D-Ala monomers that would be expected from LdcA cleavage of D44, assuming that the crosslinked D-Ala was still usable as a substrate for LdcA. Collectively, these results indicate that *A. tumefaciens* does recycle PG and thus suggested the existence of an alternative transporter of muropeptides, with a slightly altered substrate specificity to AmpG.

### Identification of the PG recycling transporter in *A. tumefaciens*

To identify the missing transporter, we took advantage of the antibiotic Fosfomycin. This inhibits the first committed step of PG synthesis, the MurA-catalysed conversion of UDP-GlcNAc into UDP-GlcNAc-enolpyruvate, which forms the substrate for the subsequent production of UDP-MurNAc by MurB[18] (Fig. 1A). This step can bypassed to some extent by the recycling of MurNAc, which confers higher intrinsic resistance to Fosfomycin[10]. We therefore used a high-throughput transposon sequencing screen (Tn-Seq) to identify genes which are required for survival under Fosfomycin treatment, reasoning that this should reveal genes required for recycling MurNAc in *A. tumefaciens*, and thus our transporter. Our Fosfomycin Tn-Seq screen revealed that recycling of MurNAc in *A. tumefaciens* likely happens as in *Pseudomonas* spp.[10–12], using the enzymes AnmK (Atu1827), MupP (Atu1614), AmgK (Atu0026) and MurU (Atu0025) to process anhydroMurNAc and convert it into UDP-MurNAc for reuse (Supplementary Data 1). In the list of genes which were statistically underrepresented in transposon insertions with 2 mg/mL Fosfomycin (Fig. 2A, Supplementary Data 1), there were two ATP-binding cassette (ABC) transporters. One consisted of the membrane-spanning and nucleotide-binding proteins of the phosphonate ABC importer *phnJK*, which is reported to form

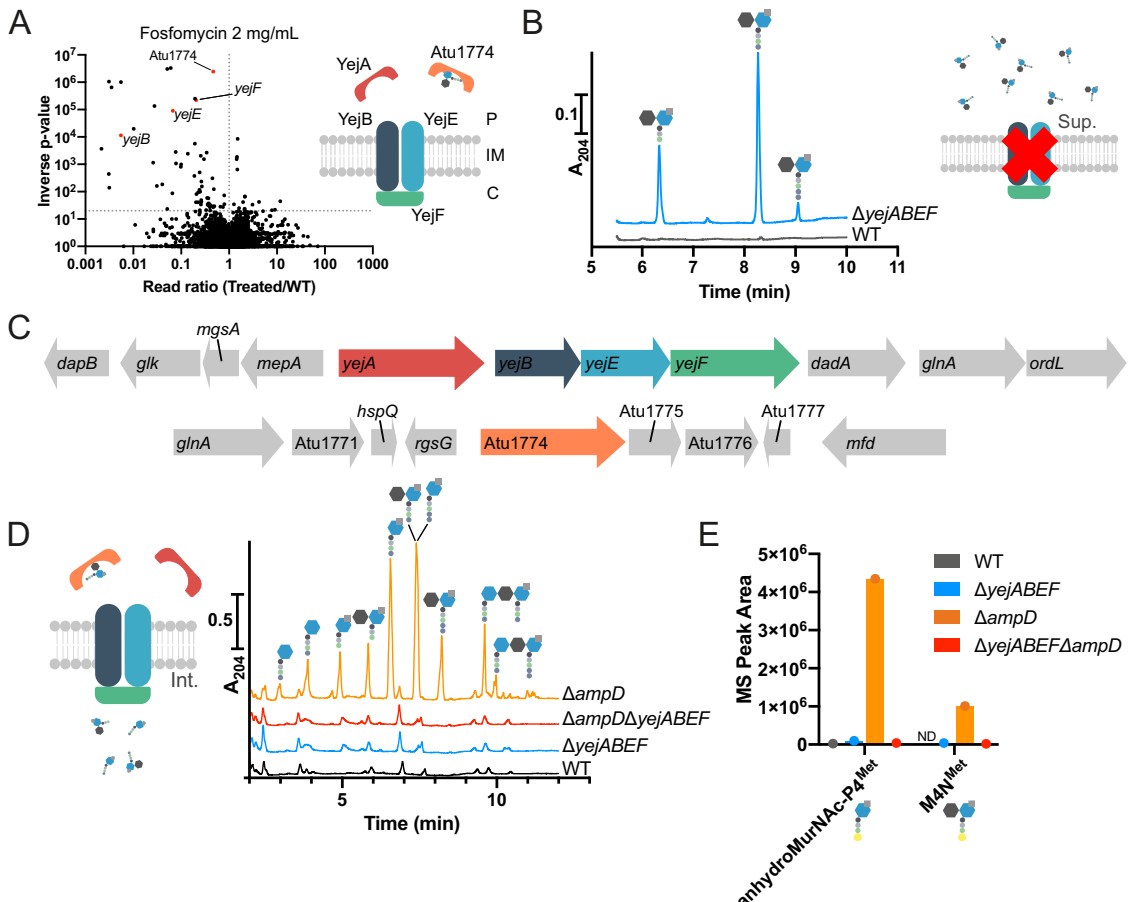

**Fig. 2 | YejBEF is the PG recycling transporter in *A. tumefaciens*. A** Volcano plot showing the ratio of Transposon sequencing (Tn-Seq) reads mapped to genes in the *A. tumefaciens* WT strain treated with 2 mg/mL Fosfomycin compared to control, plotted against inverse *p* value determined by Mann–Whitney U test (threshold $p^{-1} > 20$). Genes corresponding to *yejBEF* and Atu1774 are highlighted in the volcano plot in red; their localisation in the schematic is determined by their annotation as periplasmic substrate-binding proteins, membrane-spanning proteins or cytoplasmic ATPase. C, cytoplasm. IM, inner membrane. P, periplasm. **B** UPLC chromatogram showing levels of anhydromuropeptides released into the supernatants by *A. tumefaciens* WT and Δ*yejABEF* strains. Peaks identified using retention time and in-line LC-MS. **C** Genomic context of *yejABEF* and Atu1774. **D** Soluble PG fragments that accumulate in *A. tumefaciens* WT, Δ*yejABEF*, Δ*ampD* and Δ*yejABEF*Δ*ampD* strains detected using UPLC. **E** Quantification of D-Met containing soluble precursors in *A. tumefaciens* WT, Δ*yejABEF*, Δ*ampD* and Δ*yejABEF*Δ*ampD* strains treated with exogenous M4N[Met] by LC-MS. ND, not detected. *n* = 1 biological replicate. Source data are provided as a Source Data file.

part of the carbon-phosphorus (C-P) lyase machinery[19,20]. Since this complex mediates the cleavage of C-P bonds, such as that found in Fosfomycin, it likely confers resistance to Fosfomycin by degrading it. Indeed, degradation of Fosfomycin by Rhizobiales bacteria has been previously observed[21], and this process could explain the relatively high intrinsic Fosfomycin resistance in *A. tumefaciens*. The other ABC transporter with significantly reduced Tn insertions was a putative peptide ABC importer previously reported as YejBEF (Fig. 2A) consisting of two membrane-spanning proteins *yejB* and *yejE* (Atu0188 and Atu0189) and an ATPase *yejF* (Atu0190). Interestingly, these genes form an operon with a periplasmic SBP, *yejA* (Atu0187) which was not affected in Tn insertions under Fosfomycin treatment. Instead, our screen identified an alternative peptide-binding SBP (Atu1774) with significantly fewer Tn insertions, which showed sequence similarity to *yejA* (38.7% identity). The YejABEF ABC transporter has previously been reported as a broad-specificity peptide importer required for resistance to plant antimicrobial peptides and proper bacteroid differentiation in *Sinorhizobium meliloti*[22,23], and thus together with the SBP Atu1774, represented a good candidate for an importer of muropeptides.

To investigate whether YejABEF constitutes a novel PG recycling transporter in *A. tumefaciens*, we deleted the full operon corresponding to *yejABEF* (Atu0187-Atu0190) (Fig. 2C) and used UPLC to check the supernatant of this mutant for the accumulation of PG fragments. As would be expected, anhydromuramyl tri-, tetra- and pentapeptides accumulated in large quantities in the extracellular milieu of the Δ*yejABEF* mutant (Fig. 2B). Since deletion of *ampD* results in the accumulation of muropeptides in the cytoplasm of *A. tumefaciens*, we reasoned that deletion of the transporter in a Δ*ampD* background must necessarily prevent this. We therefore checked the cytoplasmic soluble muropeptides in a Δ*yejABEF*Δ*ampD* mutant (Fig. 2D) using UPLC. There was no cytoplasmic muropeptide accumulation in the double mutant, indicating that YejABEF is the sole transporter responsible for muropeptide recycling in *A. tumefaciens*. Further, ectopic expression of the known MFS muropeptide importer AmpG from *E. coli* restored the cytosolic accumulation of muropeptides in the Δ*yejABEF*Δ*ampD* background (Fig. S2), confirming that these transporters have the same function. Interestingly, the profile of accumulating muropeptides with AmpG expression was altered, highlighting the different specificity of these transporters (Fig. S2): there were no dimeric muropeptides, reduced levels of tri- and pentapeptide muropeptides and no GlcNAc-MurNAc muropeptides (or their products), consistent with the known specificity of AmpG[24]. Finally, we produced anhydro-disaccharide tetrapeptides with the non-canonical D-amino acid D-Met instead of D-Ala in the fourth position (M4N[Met]) as a probe for PG recycling[16]. A Δ*ampD* culture

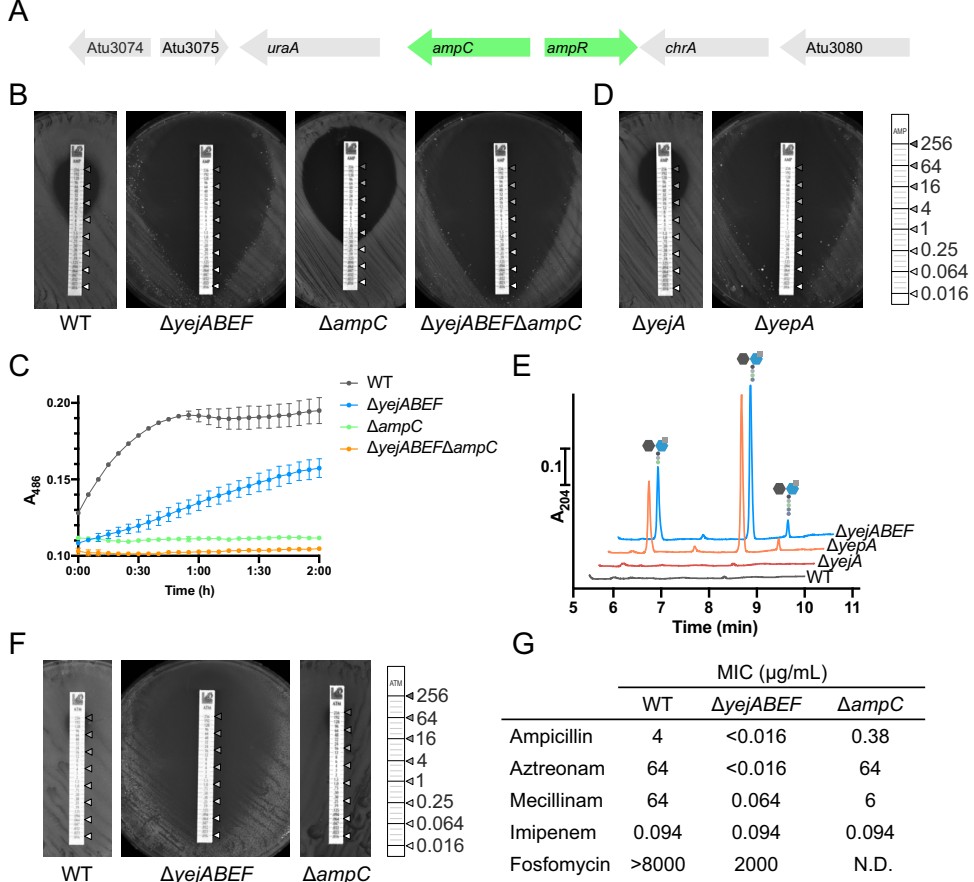

**Fig. 3 | Δ*yejABEF* is hypersensitive to Ampicillin, in a manner only partly dependent on the β-lactamase AmpC. A** Genomic context of the AmpC β-lactamase in *A. tumefaciens*, showing proximity to canonical AmpR repressor. **B** Comparison of Amp resistance of *A. tumefaciens* WT, Δ*yejABEF*, Δ*ampC* and Δ*yejABEF*Δ*ampC* strains by growth on LB plates with Ampicillin MIC test strips (μg/mL). **C** Hydrolysis of nitrocefin by lysates of *A. tumefaciens* WT, Δ*yejABEF*, Δ*ampC* and Δ*yejABEF*Δ*ampC* strains, measured by following $A_{486}$. *n* = 3 biologically independent experiments, error bars represent standard deviation from mean. Source data are provided as a Source Data file. **D** Comparison of Amp resistance of *A. tumefaciens* Δ*yejA* and Δ*yepA* (Atu1774) strains. **E** Accumulation of muropeptides in the supernatants of *A. tumefaciens* WT, Δ*yepA*, Δ*yejA* and Δ*yejABEF* strains, measured by UPLC. **F** Comparison of Aztreonam resistance of *A. tumefaciens* WT, Δ*yejABEF* and Δ*ampC* strains by growth on LB plates with Aztreonam MIC test strips. **G** Minimal inhibitory concentration (MIC) values of *A. tumefaciens* WT, Δ*yejABEF* and Δ*ampC* to β-lactams Ampicillin, Aztreonam, Mecillinam, Imipenem, and MurA-inhibitor Fosfomycin.

exogenously supplemented with M4N^Met yielded an accumulation of anhydroMurNAc-P4^Met (i.e. M4N^Met processed by NagZ) in the cytoplasm (Fig. 2E), which did not occur in the Δ*yejABEF*Δ*ampD* mutant, further confirming the requirement of YejABEF for the import of anhydromuropeptides into the cytoplasm.

### Role of PG recycling in β-lactam resistance in *A. tumefaciens*

*A. tumefaciens* encodes a canonical inducible AmpC β-lactamase system where the import of anhydromuropeptides controls β-lactamase expression through the regulator AmpR[25,26] (Fig. 3A). We therefore reasoned that the Δ*yejABEF* mutant could show decreased Ampicillin (Amp) resistance through lower β-lactamase expression. Indeed, we observed a > 250-fold decrease in Amp MIC in the transporter mutant (Fig. 3B). Consistently, hydrolysis of the chromogenic cephalosporin nitrocefin was decreased considerably in the mutant, indicating lower AmpC activity (Fig. 3C).

Interestingly, the three parts of the transporter that are required for Fosfomycin resistance form a cluster with the putative SBP YejA (Atu0187). Using RT-PCR, we confirmed that *yejA* is co-expressed with *yejBEF* on a single mRNA (Fig. S3A). As mentioned, Tn insertions in *yejA* were not significantly reduced under Fosfomycin treatment. Instead, we found that mutation of the *yejA* paralog Atu1774 conferred Fosfomycin sensitivity. To investigate the potential role of these SBPs, we generated single mutants of *yejA*

and Atu1774 and compared them with mutants of other parts of the transporter (Fig. 3C). The Amp sensitivity of the Δ*yejB*, Δ*yejE*, Δ*yejF* and ΔAtu1774 mutants recapitulated that of the entire operon deletion mutant (Fig. S3B), indicating that these components are required for the PG recycling function of the transporter, while the Δ*yejA* mutant was phenotypically identical to the WT. Additionally, we observed the accumulation of PG fragments in the supernatant of ΔAtu1774 on the same levels as for Δ*yejABEF*, while Δ*yejA* again reflected the WT (Fig. 3E). Collectively these results demonstrate that Atu1774, instead of YejA, is the SBP involved in PG recycling in *A. tumefaciens*, and so we renamed it YepA for <u>Ye</u>j-<u>p</u>eptidoglycan binding protein <u>A</u>.

Remarkably, we noticed that Δ*yejABEF* and the single mutants Δ*yejB*, Δ*yejE*, Δ*yejF* and Δ*yepA* were more Amp sensitive than Δ*ampC* (Fig. 3B), meaning that the Amp hypersensitivity of the PG recycling defective mutants was not just a result of decreased β-lactamase activity. Indeed, the nitrocefin hydrolysis assay showed that Δ*yejABEF* retained more β-lactamase activity than Δ*ampC* despite its lower Amp resistance (Fig. 3C). This was further confirmed by testing sensitivity of both mutants to aztreonam, a monobactam β-lactam antibiotic. The Δ*ampC* mutant showed an identical resistance profile to the WT, suggesting that AmpC plays no role in resistance of *A. tumefaciens* to aztreonam, while Δ*yejABEF* was once again hypersensitive (Fig. 3F).

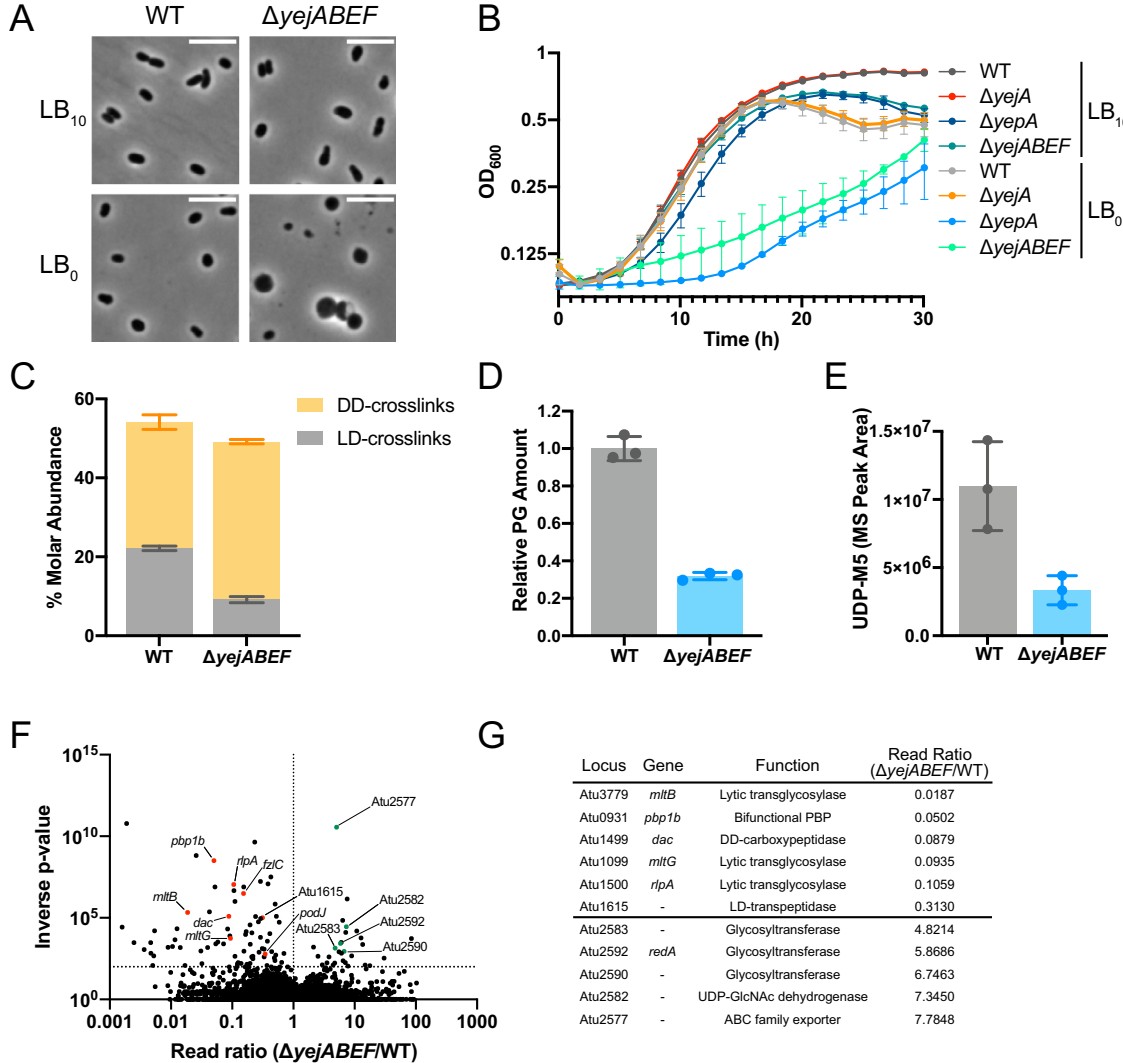

**Fig. 4 | PG recycling is required for cell wall integrity in *A. tumefaciens*.**
**A**–**E** Phenotypic analysis of *A. tumefaciens* WT and Δ*yejABEF* strains. **A** Phase contrast microscopy of WT and Δ*yejABEF* cells grown on $LB_{10}$ and $LB_0$. Scale bar length is 5 μm. Images are representative of three biologically independent replicates. **B** Growth curves of WT and Δ*yejABEF* on $LB_{10}$ and $LB_0$. $n = 3$ biologically independent experiments, error bars represent standard deviation from mean. **C** Molar abundance of DD- and LD-crosslinks in cell walls determined by UPLC analysis of WT and Δ*yejABEF* PG. **D** Relative PG amounts determined by total area under UPLC chromatograms of WT and Δ*yejABEF* PG. **E** LC-MS quantification of primary soluble

PG precursor UDP-M5 in WT and Δ*yejB*. **F** Volcano plot showing the ratio of Tn-Seq reads mapped to genes in the *A. tumefaciens* Δ*yejABEF* strain compared to WT as control. Selected synthetically detrimental (red) and synthetically beneficial (green) hits are highlighted. $p^{-1}$ value determined from Mann–Whitney U test (threshold $p^{-1} > 100$). **G** Table listing selected synthetically detrimental or beneficial hits in Δ*yejABEF* Tn-Seq discussed in the main text, with annotations. Selected hits were genes putatively related PG homeostasis or cell division with significance value above threshold. Error bars in graphs represent standard deviation from mean. Source data for **B**–**E** are provided as a Source Data file.

## PG recycling is required for cell wall integrity in *A. tumefaciens*

We therefore hypothesised that the absence of YejBEF might cause cell wall defects which result in increased sensitivity to β-lactam antibiotics. To challenge cell wall strength, we first grew the mutant and WT strains in LB with no added NaCl ($LB_0$) to see how growth would be affected by low-osmolarity conditions. The Δ*yejABEF* mutant displayed cell swelling and lysis (Fig. 4A) as well as a severe growth defect which indicated a compromised cell wall (Fig. 4B). As expected, Δ*yepA* but not Δ*yejA* recapitulated this phenotype (Fig. 4B) further strengthening the role of YepA as the PG SBP. To confirm that this effect was directly related to loss of PG recycling, we tested whether ectopic expression of *E. coli ampG* on a plasmid restores $LB_0$ growth in Δ*yejABEF* and found that it does rescue growth and morphology (Fig S4A). To investigate this effect further, we chemically analysed the cell wall of the Δ*yejABEF* mutant and WT using UPLC (Figs. 4C, D, S5). While total crosslinking was slightly decreased in the mutant (Fig. 4C), interestingly, LD-crosslinking was considerably more affected than DD-crosslinking,

which was actually slightly higher relative to the WT. Since LD-transpeptidases are not affected by most β-lactam antibiotics[17], a higher dependence of the Δ*yejABEF* mutant on PBPs for PG crosslinking could be a possible contributing factor to its decreased β-lactam tolerance. Importantly, the total amount of PG in the mutant, as determined by the area under the whole chromatogram normalised to the optical density of the culture, was decreased considerably (Fig. 4D), suggesting that PG biogenesis was compromised. Quantification of the primary soluble cytoplasmic PG precursor UDP-M5 showed a corresponding decrease (Fig. 4E), indicating that PG recycling is an important source of material for PG synthesis in *A. tumefaciens*.

To further investigate the cell wall defects of Δ*yejABEF*, we used Tn-Seq to identify genes which confer greater or reduced fitness to the mutant when interrupted by Tn insertion (Fig. 4F, G, Supplementary Data 2). Consistent with the observed defects of Δ*yejABEF* in PG synthesis and crosslinking, we found that synthetically detrimental mutants included a putative LD-transpeptidase (Atu1615), bifunctional

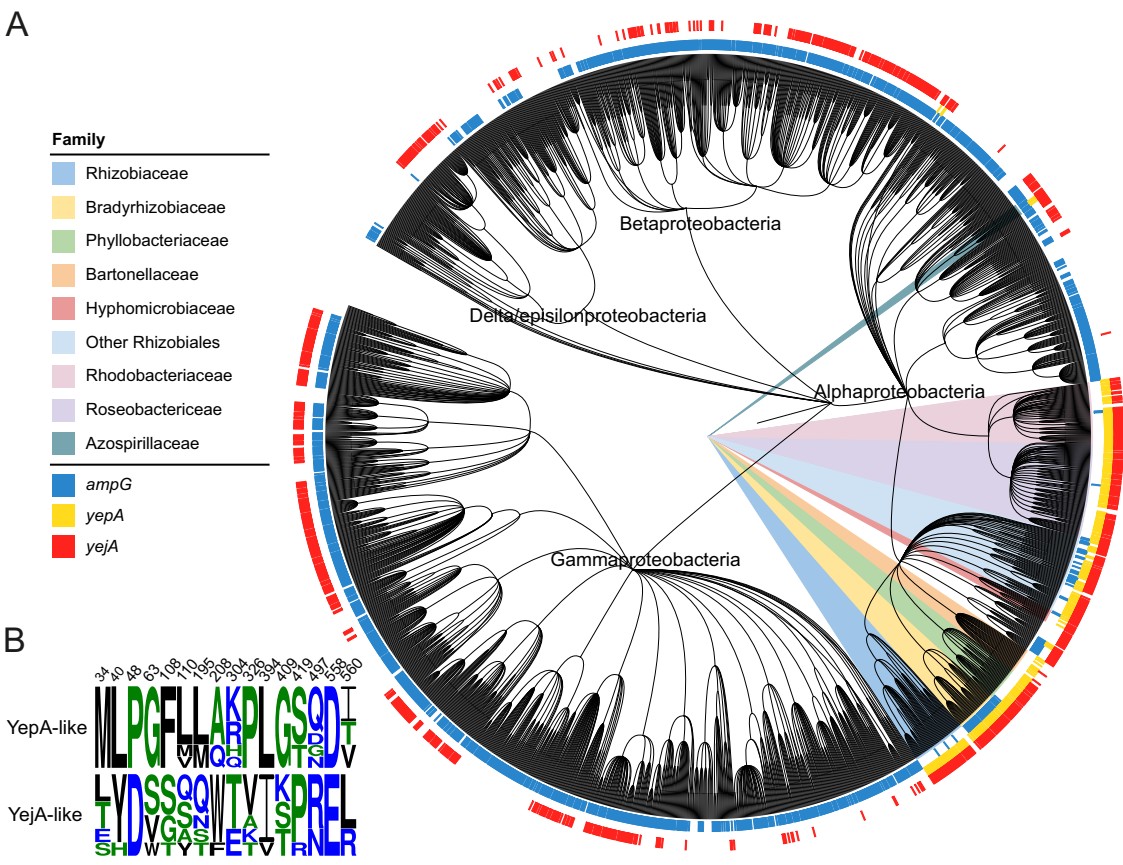

**Fig. 5 | YejABEF is widespread in the proteobacteria, but YepA is only present in the Alphaproteobacteria. A** Phylogenetic tree showing distribution of *ampG*, *yejA* and *yepA* in the Proteobacteria. **B** Molecular fingerprint residues of YepA used to differentiate between YejA and YepA.

PBP1b (Atu0931), DD-carboxypeptidase (Atu1499), and LTs MltB (Atu3779), MltG (Atu1099) and RlpA (Atu1500), suggesting that additive effects caused by the loss of these PG synthesis and remodelling proteins in combination with PG recycling results in loss of fitness or non-viability. Two other synthetically detrimental candidates were also related to cell division and cell wall synthesis, further linking YejBEF-YepA to envelope defects: PodJ (Atu0499) which denotes the old pole during division[27] and is required for chromosome segregation[28] and normal cell division[29] in *A. tumefaciens*, and the FtsZ-binding protein FzlC (Atu2824) which anchors FtsZ to the membrane in *Caulobacter crescentus* and has been linked to PG hydrolysis through synthetic lethality screens[30,31]. These results were recapitulated by a second Tn-Seq comparing insertions in the Δ*yepA* and Δ*yejA* mutants (Fig. S4B).

Conversely, mutations in several glycosyltransferases (Atu2583, Atu2590, Atu2592), a UDP-GlcNAc dehydrogenase (Atu2582) and a putative ABC exporter (Atu2577) improved fitness of the Δ*yejABEF* mutant. These genes form part of a cluster likely to be involved in the synthesis of an uncharacterised glycan, previously reported to be required for resistance to phenazines in *A. tumefaciens*[32]. Since its synthesis seems to use UDP-GlcNAc as a building block, we hypothesised that these mutations might redirect GlcNAc flux to PG synthesis, thereby improving Δ*yejABEF* fitness. However, this does not seem to be the case, as measured UDP-M5 and UDP-MurNAc levels in ΔAtu2582 and ΔAtu2582Δ*yejABEF* strains showed no differences to their WT and Δ*yejABEF* counterparts (Fig. S4C). It remains to be seen what role this polymer plays in cell wall biology in *A. tumefaciens*.

### YejBEF-YepA is conserved in the Rhizobiales and Rhodobacterales orders of the Alphaproteobacteria

Finally, to define the degree of conservation of the PG recycling function of YejBEF in bacteria, we generated a molecular fingerprint that allowed us to differentiate between YepA and YejA. Using the JDet package[33], we generated a set of specificity-determining residues from an alignment of YejA and YepA orthologs (Fig. 5B). While YejABEF is broadly conserved across the proteobacteria, YepA is exclusive to the Alphaproteobacteria (Fig. 5A), being present in almost all families in the Rhizobiales and Rhodobacterales orders, which also mostly lack AmpG. These bacteria inhabit a broad spectrum of niches including plant symbionts and pathogens (Rhizobiales), mammalian pathogens (Brucellaceae, Bartonellaceae), soil bacteria (Azospirilliceae) and photosynthetic marine bacteria (Rhodobacteriaceae) suggesting that YejBEF-YepA is not associated with one particular lifestyle. Interestingly, while most species opt either for AmpG or YejBEF-YepA, these PG transporters seem to coexist in a number of species such as in the Bradyrhizobiaceae indicating that one or both could be used under different conditions.

## Discussion

PG biosynthesis and remodelling have been the subject of decades of research, but the focus on relatively few model organisms has left the diversity of these processes poorly understood. This particularly applies to PG recycling, which has additionally suffered from a perceived lack of importance, since its interruption in the Gram-negative model organism *E. coli* has no detrimental effect to the bacterium under lab conditions[4]. Muropeptides are a unique molecular signature of bacteria which are highly immunogenic[34] and often mediate inter-species interactions[3]. Therefore, PG recycling is a factor which influences bacterial virulence and immune response activation[35–37]. In addition, the high turnover of PG during the cell cycle means that significant amounts of material would be lost if it were not recycled. Many bacteria outside the typical models lack orthologs of key PG recycling enzymes, but have others[4], suggesting there are alternative

enzymes or pathways to be discovered. For example, it was recently uncovered that both *Vibrio cholerae*[16] and *Acinetobacter baumanii*[38,39] have different functional analogs of the *E. coli* LD-carboxypeptidase LdcA with a separate but convergent evolution. We noticed that many bacteria belonging to the Rhizobiales and Rhodobacterales orders of the Alphaproteobacteria lacked an AmpG transporter despite having orthologs of the other key PG recycling enzymes AmpD and NagZ. Using the plant pathogen *A. tumefaciens* as model, we identified an alternative ABC transporter that takes over this role. *A. tumefaciens* is a fascinating model to study the growth and division of bacteria, because it exhibits an unusual asymmetric polar growth mode, and relies heavily on the typically dispensable LD-transpeptidases for PG crosslinking[40].

What happens to the PG after it is transported into the cell? An interesting difference between muropeptide recycling in *A. tumefaciens* and *E. coli* is the variety of intracellular muropeptide species which accumulate in the Δ*ampD* mutant. In *E. coli*, only the NagZ- and LdcA-processed anhydromuramyl tripeptide accumulates on significant levels[5], whereas in our *A. tumefaciens* Δ*ampD* mutant, peaks corresponding to a wide variety of anhydromuramyl tri-, tetra- and pentapeptides are present as well as dimeric muropeptides (Fig. 1B). Since direct recycling of tetrapeptide precursors is typically highly toxic to the cell[9,16], we suggest that *A. tumefaciens* does not directly reuse the PG peptide by ligation with UDP-MurNAc as in some other Gram-negative bacteria. Instead, it might simply degrade the released peptides and reincorporate the released amino acids into de novo precursor synthesis or use them as a source of C or N. This idea is supported by the lack of Mpl ligase or LD-carboxypeptidase orthologs in *A. tumefaciens*, and by the genetic context of *yejABEF* since it clusters with a putative D-alanine dehydrogenase *dadA* (Atu0192, labelled as a pseudogene due to a frameshift mutation, but this occurs after the catalytic domain suggesting that it could have retained its activity) and glutamine synthetase *glnA* (Atu0193) which uses glutamate as a substrate.

It is interesting that the PG SBP YepA is located separately from the rest of the transporter in the genome. This raises the possibility of separate transcriptional regulation of the "core" and PG-binding parts of the transporter, which would allow its PG recycling function to be regulated independently of its function as a broad-spectrum peptide importer. In addition, the presence of two SBPs competing for the same core could allow regulation of transporter function at the protein level, since higher levels of one SBP might be expected to preclude binding of the other. YepA also seems to have a conserved position beside a both a predicted cytoplasmic DsbA-like protein (Atu1775) of unknown function and the essential periplasmic polar growth protein RgsG[41], but the function of these two genes remains to be identified.

The presence of an AmpC induction system based on released muropeptides has been previously reported in *A. tumefaciens*[25], but the mechanism through which muropeptides enter the cell has not been revealed until now. As a soil bacterium, *A. tumefaciens* likely encounters β-lactam antibiotics in the environment produced by competing organisms to interact or fight for their niche[42], so the presence of an inducible β-lactamase system could be a way to counter these. However, although YejABEF does control the induction of AmpC, we showed that its effect on β-lactam sensitivity is also partly caused by cell wall defects which occur in its absence. The requirement of YejABEF for cell wall integrity and normal growth was surprising, since *E. coli* has been shown to have no physiological defects on deletion of AmpG[4]. A possible reason for this is the different ways in which these bacteria grow: while *E. coli* elongates all along the cell[43], *A. tumefaciens* elongates from one pole[40]. Integrity of the newly synthesised PG is perhaps more susceptible to a lowered flux of material when synthesis is concentrated on a single area.

Our Tn-Seq screen in Δ*yejABEF* revealed synthetically lethal cell wall-related genes including a bifunctional PBP, an LD-transpeptidase,

and three LTs. Insertions in the synthetic PBP and LD-transpeptidase could result in lethality due to additive effects on top of the synthesis defects that already occur. Mutation of LTs is known to cause periplasmic stress due to periplasmic crowding caused by the accumulation of PG chains[44]. Interruption of PG recycling could exacerbate periplasmic crowding, since we observed PG fragments accumulating in large amounts in the supernatant of Δ*yejABEF*. That we observe chains accumulating in the cytoplasm of Δ*ampD* but not in the supernatant of Δ*yejABEF* suggests they could be too large to escape the outer membrane, accumulating in the periplasm and thereby causing osmotic stress.

Our Tn-Seq screen also revealed that Tn insertions in a gene cluster likely responsible for the synthesis of an uncharacterised glycan alleviated the fitness cost of deleting Δ*yejABEF*. While this could be consistent with the idea of alleviating periplasmic crowding, *A. tumefaciens* also produces osmoregulated cyclic β-glucans[45], which did not show up in our screen, meaning this is likely not the case. This polymer has previously been reported as required for tolerance of the phenazine pyocyanin in *A. tumefaciens*[32], suggesting it could play some role in oxidative stress metabolism. Cell wall defects have previously been associated with an overproduction of hydroxyl radicals[46], and the potential role of YejABEF in periplasmic oxidative stress is a subject for future research.

YejABEF has previously been reported as important for proper bacteroid differentiation and resistance to the plant antimicrobial peptide NCR247 in *S. meliloti*[23,47]. Though this was thought to be due to its activity as a promiscuous peptide importer, we suggest that it actually comes from its function as a PG transporter, since AmpD is also required for NCR247 tolerance and the non-PG SBP YejA is not[47]. Indeed, the bacteroids formed by *S. meliloti* Δ*yejE* and Δ*yejF* mutants display an aberrant, swollen morphology[23] indicating that PG recycling could be necessary to maintain cell wall integrity in *S. meliloti* during bacteroid differentiation. *A. tumefaciens*, along with *Brucella abortus* and *Bartonella henselae*, is an intracellular pathogen, and so "hiding" its released PG from the host could be a strategy to avoid immune detection since immune receptors for PG are located intracellularly. The presence of the transporter in many other Alphaproteobacterial species, including some which share an intracellular or host-associated niche, makes it an attractive topic for further study.

The question of why *A. tumefaciens* uses an ABC transporter to recycle PG instead of a PMF-dependent MFS transporter such as AmpG remains. The use of different energy sources could mean that PG recycling is required by *A. tumefaciens* under a specific condition where ATP is available as an energy source but membrane polarisation is affected. However, *yepA* does not seem to be limited to any one particular niche, being present in diverse plant and animal pathogens and symbionts and also environmental and marine bacteria (Fig. 5). The Rhizobiales are known to be particularly rich in ABC transporters, with 146 full ABC importers present in *Sinorhizobium meliloti*[48,49]. One possible reason for this is that it could facilitate a common mechanism of regulation of transport. Inactivation of ABC transporters on a cellular scale has been identified previously in *Rhizobium leguminosarum* in response to sensing of global cellular nitrogen availability[50–52] or glutathione levels[53], through a putatively post-translational mechanism which remains to be elucidated[50,52]. Therefore, using an ABC transporter for PG recycling could allow the cell to coordinate the regulation of this process with the regulation of acquisition of other nutrients and energy sources, perhaps in synchronisation with the cell cycle since glutathione levels are known to fluctuate during the cell cycle in the related Alphaproteobacterium *Caulobacter crescentus*[54].

## Methods

### Strains and growth conditions

Strains used in this study are listed in Table S1. *A. tumefaciens* strains were grown in LB, ATGN or ATSN (prepared as ref. 55 but without the

addition of $FeSO_4$ to avoid precipitation as its absence did not affect growth) at 30 °C unless stated otherwise. *E. coli* strains were grown in LB at 37 °C. All *A. tumefaciens* mutants are derivatives of the *Agrobacterium fabrum* strain C58. When required, antibiotics were added to growth media: Kanamycin (50 µg/mL *E. coli*, 300 µg/mL *A. tumefaciens*) or Amp (100 µg/mL). Growth curves were done using a microplate reader (Eon BioTek, with the BioTek Gen5 3.09 software) to measure $OD_{600}$ of 200 µL cultures in a 96-well microplate over 28 h at 5 min intervals. Three technical replicates of three biological replicates (i.e. separate parental cultures) were used per condition. Initial cultures were prepared by adjusting exponentially growing cultures to $OD_{600}$ 0.01.

### Construction of plasmids and mutants
Expression and allelic exchange vectors were constructed using the primers outlined in Table S2. Allelic exchange vectors for *A. tumefaciens* were constructed using isothermal assembly of flanking regions with the suicide vector pNPTS139; mutagenesis was carried out as described[56]. Protein expression vectors were constructed by PCR of the region of interest with addition of restriction sites by primers, then restriction digest and ligation before transformation into *E. coli* DH5α.

### Antibiotic resistance determination
Resistance to various antibiotics was determined on agar using MIC Test Strips (Liofilchem). Exponential cultures were adjusted to $OD_{600}$ 0.1 in PBS before being spread across an LB agar plate three times using a sterile cotton swab; the plate was rotated 60° between each spreading. The strip was applied once the plate had dried fully.

### Nitrocefin hydrolysis assay
Overnight cultures of *A. tumefaciens* strains were diluted 1:3 in fresh LB with 50 µg/mL Amp added (to induce *ampC* expression) and grown for 1 h. 1 mL of culture normalised to $OD_{600}$ 0.4 was prepared which was pelleted and washed in 1 mL of PBS. The final pellet was then resuspended in 1 mL lysis buffer (PBS with 0.2% Triton X-100, 50 µg/mL lysozyme, and 2 mM EDTA) and incubated on ice for 30 min to lyse the cells. 200 µL reactions were set up containing 100 µL lysate, 96 µL PBS and 4 µL 500 µg/mL nitrocefin (Oxoid) and the reaction was followed by monitoring $A_{486}$ for 2 h in a microplate reader (Eon Biotek) with readings every 5 min.

### Soluble muropeptide analysis
To determine the presence and levels of intra- and extracellular soluble muropeptides, bacteria were grown until late exponential phase (roughly $OD_{600}$ 0.7) in minimal media before being cooled on ice for 10 min and normalised to the same $OD_{600}$. Cells were then harvested by centrifugation at $10,000 \times g$ for 2 min. The supernatant was decanted and passed through a 0.22 µm filter, then boiled for 10 min, centrifuged at maximum speed in a benchtop centrifuge for 10 min to remove precipitated proteins and concentrated 20 times in a Speed-Vac system. Meanwhile, the cell pellet was washed three times in ice-cold 0.9% NaCl, resuspended in 0.9% NaCl so that the cells are 20 times concentrated and boiled for 10 min before centrifugation at maximum speed in a benchtop centrifuge for 10 min to remove the proteins and insoluble fraction.

Detection and chemical characterisation of the soluble muropeptides was carried out using an Acquity H-Class UPLC (Waters) coupled to a Xevo G2/XS QTOF mass spectrometer. Chromatographic separation was achieved using an Acquity UPLC BEH C18 column (Waters) maintained at 45 °C. A linear gradient of 0.1% formic acid in water to 0.1% formic acid in acetonitrile was used for elution. UV detection was performed at 204 nm. The QTOF instrument was operated in positive ion mode, with data collection performed in untargeted $MS^e$ mode. The parameters were set as follows: capillary voltage

3.0 kV, source temperature 120 °C, desolvation temperature 350 °C, sample cone voltage 40 V, cone gas flow 100 L h$^{-1}$ and desolvation gas flow 500 L h$^{-1}$. Data acquisition and processing was performed using the UNIFI software (Waters). A compound library of expected muropeptides was built in UNIFI by drawing the structures in ChemSketch (http://www.acdlabs.com). This library was used for automated processing of the data and quantification by integration of the peaks from extracted ion chromatograms.

### Peptidoglycan analysis
Peptidoglycan isolation and analysis were carried out as described previously[57]. Briefly, 10 mL stationary phase cultures were normalised to the same $OD_{600}$, pelleted, resuspended in 5% SDS and boiled for 2 h with stirring. The obtained sacculi were washed repeatedly in water to remove the SDS before analysis; pelleting was performed by ultracentrifugation in an Optima MAX-TL benchtop ultracentrifuge at $150,000 \times g$. The final pellet was resuspended in water and digested overnight with muramidase (80 µg/mL) before adjustment to pH 9 with addition of 0.5 M sodium borate, reduction with addition of 10 mg/mL sodium borohydride for 20 min at RT and adjustment to pH 3.5 by addition of orthophosphoric acid.

Liquid chromatography was carried out using an Acquity H-Class UPLC system (Waters) equipped with an Acquity UPLC BEH C18 column (2.1 mm × 150 mm, 130 Å pore size and 1.7 µm particle size, Waters). Chromatographic separation was achieved using a linear gradient from 0.1% formic acid in water to 0.1% formic acid in acetonitrile. Peaks were identified using in-line LC-MS and assigning peak identities using retention time. The relative amount of each muropeptide was calculated by integrating all the peaks and dividing each peak area by the total area of the chromatogram, while the total area of the chromatogram was taken as the relative density of PG per sample and expressed relative to the WT.

### M4N$^{Met}$ incorporation experiment
M4N was prepared by digesting purified *V. cholerae* sacculi using the lytic transglycosylase MltA. An in vitro Ldt-mediated exchange reaction with purified LdtA was used to swap the D-Ala in position 4 with D-Met to produce M4N$^{Met}$ as described[58]. 10 mL of exponentially growing *A. tumefaciens* cultures was pelleted and resuspended in 200 µL fresh medium, and ~15 µg M4N$^{Met}$ was added to each. The cultures were then grown for a further hour and prepared for soluble precursor analysis as described above.

### Bioinformatic analyses
Sequences of SBPs corresponding to YejA and YepA were identified in representative species spanning the Rhizobiales and Rhodobacterales orders of the Alphaproteobacteria: *A. tumefaciens* (UniProt YejA: A9CKL4, YepA: A9CIN5), *Sinorhizobium meliloti* (YejA: Q92T30, YepA: Q92PT7), *Bartonella henselae* (YepA only: A0A0H3LXD4), *Brucella melitensis* (YejA: Q8YEE4, YepA: Q8YC41), *Ochrobactrum anthropi* (YejA: A6WUU2, YepA: A6X540), *Dinoroseobacter shibae* (YejA: A8LQB5, YepA: A8LK88) and *Phaeobacter inhibens* (YejA: I7EQS1, YepA: I7EWM7). SBPs were identified by phenotype for *D. shibae* and *P. inhibens* (using FitnessBrowser[59] to look for the SBP with the expected β-lactam and Fosfomycin sensitivity phenotypes) and eggNOG 5.0[60] using fine-grained orthologs of Atu1774 and Atu0187 for the others. Sequences were aligned using MUSCLE[61] and the JDet package[33] was used to find specificity-determining positions (SDPs), which were visualised using WebLogo[62]. To produce a phylogenetic tree, all YejA orthologs were downloaded from OrthoDB v10 (group 231800at1224)[63], aligned using MUSCLE v5 and the previously determined SDPs were used to label each as either YejA or YepA. AmpG orthologs from OrthoDB v10 (group 542870at1224) were filtered by BLASTP against *E. coli* AmpG (UniProt P0AE16) using an E value cutoff of $1 \times 10^{-25}$. PhyloT was used to produce a phylogenetic tree of all

species on OrthoDB and orthologs of YejA, YepA and AmpG were mapped against this. The final tree was visualised using iTOL[64].

### RT-PCR of *yejABEF* cluster

All kits were used as per the manufacturer's instructions. Total RNA was extracted from *A. tumefaciens* C58 WT using the RNeasy Plus Mini Kit (QIAGEN), DNA was degraded using the TURBO DNA-free kit (Thermo) and cDNA was synthesised using the Maxima Reverse Transcriptase (Thermo). PCR was performed with primers FCP6349-6357 using the VeriFi polymerase (PCRBIO), with a 64 °C annealing temperature and 1.5 min elongation time.

### Transposon sequencing

For identification of conditionally essential genes, Tn-Seq was performed broadly as described elsewhere[65], adapted for *A. tumefaciens*. $3-5 \times 10^5$ transposon mutants were generated per library by conjugation of *A. tumefaciens* C58 with *E. coli* SM10λ-pir carrying the mariner transposon donor plasmid pSC189[66]. Mutant libraries were selected on LB plates with 500 µg/mL kanamycin (to select for the transposon), 25 µg/mL streptomycin (to remove the donor *E. coli*) and 2 mg/mL Fosfomycin where appropriate. The libraries were collected, genomic DNA extracted, and pooled genomic DNA fragments were sequenced using a MiSeq system (Illumina). Insertion sites were identified and statistical representation of transposon insertions determined using the ConArtist pipeline as described in ref. 67.

### Reporting summary

Further information on research design is available in the Nature Portfolio Reporting Summary linked to this article.

## Data availability

Raw sequencing reads for Tn-Seq experiments are available at the NCBI Sequence Read Archive (BioProjects PRJNA869898 and PRJNA869899). Sequences for YejA and AmpG orthologs were obtained from OrthoDB v10[63], groups 231800at1224 for YejA and 542870at1224 for AmpG. Reference AmpG and SBP sequences were obtained from UniProt under accession codes P0AE16 for AmpG and as follows for SBPs: *A. tumefaciens* (YejA: A9CKL4, YepA: A9CIN5), *Sinorhizobium meliloti* (YejA: Q92T30, YepA: Q92PT7), *Bartonella henselae* (YepA only: A0A0H3LXD4), *Brucella melitensis* (YejA: Q8YEE4, YepA: Q8YC41), *Ochrobactrum anthropi* (YejA: A6WUU2, YepA: A6X540), *Dinoroseobacter shibae* (YejA: A8LQB5, YepA: A8LK88) and *Phaeobacter inhibens* (YejA: I7EQS1, YepA: I7EWM7). All other data generated or analysed during this study are included in this published article (and its supplementary information files). Source data are provided with this paper.

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

## Acknowledgements

We thank all members of the Cava lab for helpful comments and suggestions. We are grateful to Aysha Chowdury for technical assistance in construction of the ΔAtu2582 and ΔAtu2582Δ*yejABEF* mutants. Research in the Cava lab is supported by the Swedish Research Council, the Laboratory for Molecular Infection Medicine Sweden (MIMS), Umeå University, the Knut and Alice Wallenberg Foundation (KAW) and the Kempe Foundation.

## Author contributions

M.C.G. and F.C. conceived the study, designed the experiments, interpreted the data and wrote the paper. M.C.G. performed experiments.

## Funding

## Competing interests

The authors declare no competing interests.
