## [Peer Review File · Nature Communications]

Peptidoglycan recycling mediated by an ABC transporter in the plant pathogen *Agrobacterium tumefaciens*Reviewer #1 (Remarks to the Author):

The ms by Gilmore and Cava reports on the peptidoglycan recycling metabolism and on the identification of a novel transport system required for muropeptide uptake in *Agrobacterium tumefaciens*. This organism belongs to the alpha-proteobacteria, a neglected group of bacteria in respect to cell wall turnover and recycling - nevertheless worth studying due to their unique polar mode of growth and their medically and biologically important members. The authors noticed that alpha-proteobacteria lack an ortholog of the well-known peptidoglycan recycling transporter AmpG, despite that cytoplasmic peptidoglycan recycling enzymes are present. The authors first showed that indeed peptidoglycan recycling occurs in A.t. and then they used Tn-Seq to identify genes required for survival of fosfomycin treatment, i.e. are unable to bypass the fosfomycin target reaction of peptidoglycan synthesis. Thereby they identified the ABC transporter YejBEF, including the solute binding protein YepA, and subsequently showed that it imports (anhydro-) muropeptides and governs expression of AmpC beta-lactamase, thus takes over both function of the AmpG MFS transporters in other bacteria, and in addition appears to be crucial for cell viability. To my knowledge, this is the first report of an ABC transporter transporting peptidoglycan sugar peptides in bacteria, which resembles in a way the Opp/MppA peptide recycling system of *E. coli*. In both systems an ABC core transporter recruits an alternative solute binding protein to confer specificity to peptidoglycan fragments.

The ms is well structured, concise and clearly written. It is technically sound and provides an advancement to the scientific field. Still, I have some issues that should be addressed in a revision of the ms.

A major concern is the unclear assignment and insufficient presentation of the muropeptides in the LC-MS experiments shown. In Figure 1B a "base peak chromatogram" is shown, the identification of muropeptides by their exact masses according to table S3, however, requires extracted ion chromatograms that should be shown. In Suppl.Fig. 1 the "Total Ion Current Intensity" is shown: why TIC and not BPC ? Again EICs are missing. All the other chromatograms apparently are UV traces (Fig. 2B, 2D, 3E). Here it is totally unclear, if the assignment of a compound occurred solely on the basis of retention time ? Or where the peak fractions collected and re-measured by LC-MS ?

Another point of concern is the conclusion that PG recycling is vital for cell wall integrity. The transporter was previously identified as required for antimicrobial peptide resistance. So it cannot be ruled out that some antimicrobial peptide is produced by A.t. and the transporter is required for resistance ?

Figure 1: The overview scheme is oversimplified, to my impression. In Fig. 1B and in all the other Figure containing chromatograms, the muropeptide symbols, particularly the amino acid symbols are very hard to read. I would recommend to use letters and numbers (e.g. M4; aM3..) instead of the cartoon labels.

L85: The authors first aimed to confirm that peptidoglycan recycling occurs in A.t. using muropeptide LC-MS analyses of extracts of mutant cells. Although accumulation of muropeptides indicates that AmpD and NagZ of A.t. have orthologous functions to the *E. coli* counterparts, the recycling fragments are somewhat different and thus I do not agree with the conclusion that they "play the same respective roles in PG recycling". In *E. coli* and other Gram-negative bacteria no accumulation of MurNAc-peptides, GlcNAc-MurNAc, and trisaccharides occur, thus in A.t. different recycling products occur. If (cf. l145), yejBEF-YepA is the sole transporter for muropeptides it should thus differ in substrate specificity from AmpG, which is specific for anhydro-muropeptides. These differences should be evaluated and reported. The differences in specificity should be easily retrieved from existing data: e.g. in L146ff the functional complementation by ectopic ampG expression was mentioned. These data (Suppl. Fig. 2) should allow conclusion to be drawn about the specificity of the YejBEF-YepA transporter.

In Figs. 2B and 2D only the UV traces of a UPLC separation were shown. Supposedly, no standard muropeptides were used for identification. So, how the compounds were identified/ how were the masses of the compounds determined?

L96: The conclusion that chains rather than crosslinked dimers accumulate cannot be drawn, since the occurrence of 3-3 dimers cannot be ruled out. For a clearly proof of identity, the fragments could be cleave by an amidase (rather than with LdcA) that would produces either peptide monomers or peptide dimers.

Fig. 4C: it is unclear how the amounts of D,D and L,D crosslinks were determined - no data are shown ?

I am missing a discussion about the peculiarity of PG recycling during apical growth. Is this possibly the reason for being vital for cell wall integrity ?

Reviewer #2 (Remarks to the Author):

Bacteria of the Rhizobiales/Rhodobacterales orders of Alphaproteobacteria lack the model system muropeptide transporter AmpG, despite having other key PG recycling enzymes. The authors use a drug resistance screen to identify an alternative transporter system, YejBEF-YepA, that plays this role in the Rhizobiales phytopathogen *Agrobacterium tumefaciens*, and absence of YejBEF-YepA causes severe cell wall and growth defects. Thus, PG recycling is essential for cell wall integrity in *A. tumefaciens*. This result up-ends previous studies in model Gram negative bacteria, such as *E.coli*, where deleting genes required for PG recycling has no effect on cell growth under lab conditions. The data underscore the need to further study such pathways in important non-model systems that act in important medical and plant specific pathogens.

Most experiments are of quality and support the conclusions drawn. However, clarifications and additions to the text and figures are warranted as described below.

1.The manuscript spends a lot of words on speculations, especially in the Discussion. Instead, the results would benefit from expansion while the Discussion is reduced.

2. The figures throughout showing ampicillin MIC test strips are not clear. The strips themselves should be removed, as they will not be readable when such figures are reduced during potential publication. The different degrees of bacterial growth on the plates are obviously different. It is not clear what the strips were meant to show relative to the growth patterns; this needs to be explained in more detail. The authors should design an overlay scale (to place on top of the images of bacterial growth on plates) that is readable and interpretable?

3.Figure 1A PG recycling pathway needs to be expanded. Where does Fosfomycin act in the pathways shown? Where does the MurA-catalysed conversion of UDP-GlcNAc into UDP-1-carboxyvinyl-GlcNAc occur? Where is substrate for the subsequent production of UDP-MurNAc by MurB? Where does the recycling of MurNAc occur? These edits to the figure would help the reader better understand lines 101-106.

4.The diagram in Figure 2A is similarly cryptic. The legend should better justify why the red, orange, dark blue, light blue, green proteins are placed in the cytoplasm, membrane, or periplasm.

5. There is no legend for Fig.2E. Fig. 2E itself seems to be missing information as only black and orange data are shown and no blue or red data are shown.

6.Lines 162-163 say there is " > 250 fold decrease in Amp MIC in the transport mutant" (Fig. 3B). How can we see this from the plates shown, especially when one cannot read the strips? Please clarify.

7. Lines 171-173 say that Fig. 3C will compare mutants of *yjA* and *Atu1774*, but these mutants are not found in this panel or any other panels of Fig. 3? Line 178 also says these same mutants will be compared in Fig.3E, but they are not shown on this panel?

The authors make a conclusion about this data (lines 179-181) but where are the data?

8. Basically, it is confusing to the reader when the authors refer to the wrong figures. Please check throughout. Additionally, lines 182 -183 refer to Fig. 3B, but presumably the authors mean to say Supp. Fig.3B?

9. Lines 199-200. The authors should combine Fig.4A and Supp. Fig. 4A to be able to compare all the data on a single graph.

10. Lines 214-235. It is not clear why the authors choose to mention particular genes of interest. Logic of choices should be justified, with cross reference to the read ratios in panel F.

11. Lines 311-315. Speculation about *A. tumefaciens* encountering antibiotics in the soil microenvironment seems unwarranted, unless all our water is contaminated?

12. Lines 343-345. Bacteroids are extremely large and enclosed in plant membranes. It is not appropriate to compare their morphologies with those of single mutant cells in *A. tumefaciens* that are simply swollen due to defects in PG.

13. The rest of this paragraph to line 354 is highly speculative.

Reviewer #3 (Remarks to the Author):

Gilmore and Cava report on the identification of an ABC transporter responsible for peptidoglycan recycling in *Agrobacterium tumefaciens*. The finding is original and unexpected because muropeptide recycling in well-studied Gram-negative model bacteria depends on an entirely different transporter, namely the proton-driven MFS transporter AmpG. Since *Agrobacterium* and related bacteria do not encode an AmpG homolog, the authors established a clever Tn-Seq screen to identify the responsible transport system in *Agrobacterium*. They show that – in contrast to *E. coli* - the transporter is vital.

The story is straightforward, the findings are novel, and the manuscript is well written. I only have a couple of comments and questions.

Questions and comments:

1. Fig. 1B: I did not understand why only a small and a single peak appears in the *nagZ* mutant whereas multiple peaks appear in the *ampD* mutant. Wouldn't one expect the opposite since *NagZ* acts upstream of *AmpD* (Fig. 1A)? Please clarify.

2. Is the *yejABEF* mutant sensitive to antibiotics other than beta-lactams or to membrane-damaging agents? Does the mutant have any other growth defects (except for the ones shown in Fig. 4)?

3. Fig. 2C: Are the two gene clusters located on the same replicon (circular or linear chromosome)?

4. Fig. 4A: Add length of the scale bar in the figure or legend.

5. Lines 298/299: Do transcriptomics data in public databases tell us anything about the expression of these genes?

6. Likewise for bacteria that contain both transport system (e.g. *Bradyrhizobium*): Do publicly available transcriptomics data show whether both systems are (differentially) expressed under laboratory conditions or in the plant?

REVIEWER COMMENTS

Reviewer #1 (Remarks to the Author):

The ms by Gilmore and Cava reports on the peptidoglycan recycling metabolism and on the identification of a novel transport system required for muropeptide uptake in *Agrobacterium tumefaciens*. This organism belongs to the alpha-proteobacteria, a neglected group of bacteria in respect to cell wall turnover and recycling - nevertheless worth studying due to their unique polar mode of growth and their medically and biologically important members. The authors noticed that alpha-proteobacteria lack an ortholog of the well-known peptidoglycan recycling transporter AmpG, despite that cytoplasmic peptidoglycan recycling enzymes are present. The authors first showed that indeed peptidoglycan recycling occurs in A.t. and then they used Tn-Seq to identify genes required for survival of fosfomycin treatment, i.e. are unable to bypass the fosfomycin target reaction of peptidoglycan synthesis. Thereby they identified the ABC transporter YejBEF, including the solute binding protein YepA, and subsequently showed that it imports (anhydro-) muropeptides and governs expression of AmpC beta-lactamase, thus takes over both function of the AmpG MFS transporters in other bacteria, and in addition appears to be crucial for cell viability. To my knowledge, this is the first report of an ABC transporter transporting peptidoglycan sugar peptides in bacteria, which resembles in a way the Opp/MppA peptide recycling system of *E. coli*. In both systems an ABC core transporter recruits an alternative solute binding protein to confer specificity to peptidoglycan fragments.

The ms is well structured, concise and clearly written. It is technically sound and provides an advancement to the scientific field. Still, I have some issues that should be addressed in a revision of the ms.

We thank the reviewer for their detailed read of our manuscript and their positive comments.

A major concern is the unclear assignment and insufficient presentation of the muropeptides in the LC-MS experiments shown. In Figure 1B a "base peak chromatogram" is shown, the identification of muropeptides by their exact masses according to table S3, however, requires extracted ion chromatograms that should be shown. In Suppl.Fig. 1 the "Total Ion Current Intensity" is shown: why TIC and not BPC ? Again EICs are missing. All the other chromatograms apparently are UV traces (Fig. 2B, 2D, 3E). Here it is totally unclear, if the assignment of a compound occurred solely on the basis of retention time ? Or where the peak fractions collected are re-measured by LC-MS ?

We agree with the reviewer that the alternative use of TIC and BPI traces is confusing. This was done purely for aesthetic reasons, and we have altered all figures to use the respective TIC traces instead.

Regarding the identification of muropeptides, we have added EICs (for the *ampD/nagZ* mutants in Fig. 1 and Fig S1 *ampD* ± *ldcA* expression strains) as part of a supplementary figure (Fig S1B and S1C). Where UV traces are shown, peak identification was done using in-line UPLC-MS, since the masses and retention times are the same as those already shown via EICs. We have added a sentence to the respective figure legends to clarify this. For the added cell wall analysis (see below), we have reported the masses in a table along with the probable structure, as is common practice for these analyses (e.g., (1-3)).

Another point of concern is the conclusion that PG recycling is vital for cell wall integrity. The transporter was previously identified as required for antimicrobial peptide resistance. So is cannot be ruled out that some antimicrobial peptide is produced by A.t. and the transporter is required for resistance ?

The reviewer raises an interesting question. We believe that this is not the case, because ectopic expression of *E. coli ampG* actually restores cell wall integrity. This means that it almost certainly is directly related to PG recycling due to the strict substrate specificity of AmpG (4). We have added this data to supplementary Figure 4 (A and B panels) in the form of an LB₀ growth curve and microscopy images, which show convincing morphological and partial growth complementation. That growth complementation is partial could be due to the non-native nature of AmpG in *A. tumefaciens* resulting in reduced activity, or the difference in "cargo" between the two transporters for which we refer to our answer to the reviewer's question two after this one. We have added a sentence detailing this effect in the results section (lines 213-216). We thank the reviewer for their interesting idea, and believe this additional data strengthens the conclusions of the paper.

Figure 1: The overview scheme is oversimplified, to my impression. In Fig. 1B and in all the other Figure containing chromatograms, the muropeptide symbols, particularly the amino acid symbols are very hard to read. I would recommend to use letters and numbers (e.g. M4; aM3..) instead of the cartoon labels.

We thank the reviewer for their feedback, and have expanded the overview scheme to include the *Pseudomonas* MurNAc recycling pathway discussed in the text, and also the point where Fosfomycin

acts. In this case, we favour the use of the schematic or cartoon muropeptides because the literal processing of the muropeptides is quite visually striking, particularly for those readers who aren't familiar with peptidoglycan metabolism. However, we have changed the colour of the mDAP amino acid to be more distinctive. Since it sits in the middle of the peptide chain, we believe this helps the brain to process the chain composition.

L85: The authors first aimed to confirm that peptidoglycan recycling occurs in A.t. using muropeptide LC-MS analyses of extracts of mutant cells. Although accumulation of muropeptides indicates that AmpD and NagZ of A.t. have orthologous functions to the E. coli counterparts, the recycling fragments are somewhat different and thus I do not agree with the conclusion that they "play the same respective roles in PG recycling". In E. coli and other Gram-negative bacteria no accumulation of MurNAc-peptides, GlcNAc-MurNAc, and trisaccharides occur, thus in A.t. different recycling products occur. If (cf. L145), yejBEF-YepA is the sole transporter for muropeptides it should thus differ in substrate specificity from AmpG, which is specific for anhydro-muropeptides. These differences should be evaluated and reported. The differences in specificity should be easily retrieved from existing data: e.g. in L146ff the functional complementation by ectopic ampG expression was mentioned. These data (Suppl. Fig. 2) should allow conclusion to be drawn about the specificity of the YejBEF-YepA transporter.

We agree that it makes sense to report the altered substrate specificity of YejBEF-YepA and have included both a mention after the discussion of the dimeric muropeptides (line 106), as well as a short paragraph discussing the alternative accumulation in the $\Delta ampD$ pSRK::ampG expression experiment (lines 156-160):

"Interestingly, the profile of accumulating muropeptides with AmpG expression was altered, highlighting the different specificity of these transporters (Fig. S2): there were no dimeric muropeptides, reduced levels of tri- and pentapeptide muropeptides and no GlcNAc-MurNAc (i.e., not anhydroMurNAc) muropeptides (or their products), consistent with the known specificity of AmpG (4)."

In Figs. 2B and 2D only the UV traces of a UPLC separation were shown. Supposably, no standard muropeptides were used for identification. So, how the compounds were identified/ how were the masses of the compounds determined?

As described in our previous answer, peak identification was done using in-line LC-MS where the analyte coming from the UV detector goes directly into an ESI-QTOF MS, allowing us to easily correlate the two. We have added a line to figure legend and methods (line 481-482) to explain this.

L96: The conclusion that chains rather than crosslinked dimers accumulate cannot be drawn, since the occurrence of 3-3 dimers cannot be ruled out. For a clearly proof of identity, the fragments could be cleave by an amidase (rather than with LdcA) that would produce either peptide monomers or peptide dimers.

We disagree with the reviewer that the occurrence of 3-3 dimers cannot be ruled out. In LD- or 3-3-crosslinked muropeptides, the donor muropeptide within the dimer must necessarily be a tripeptide, and the final identity of the crosslinked dimer (D33, D34, D35) depends on the acceptor for the reaction (5) but cannot be D44. That the most prominent dimer peak changes from being an anhydrotrisaccharide-ditetrapeptide to an anhydrotrisaccharide-ditriptide upon digestion with the LD-carboxypeptidase LdcA is only compatible with a linear (non-crosslinked) M4-M4N, as digestion of a DD-crosslinked D44N by LdcA would produce M3 + M4-D-Ala instead). We have expanded this section in the text to clarify this (lines 97-105).

Fig. 4C: it is unclear how the amounts of D,D and L,D crosslinks were determined - no data are shown ?

We thank the reviewer for pointing this out. The amounts of L,D- and D,D-crosslinks were determined from a full UPLC cell wall analysis taking into account the relative abundance of each muropeptide species (6). The chromatograms corresponding to the WT and $\Delta yejB$, showing LD- (*) and DD-crosslinked peaks as well as the quantification from where this graph is derived are shown below :

While revising the manuscript, we decided that it would make for a better paper and more logical flow for the reader to have this data for the full transporter mutant $\Delta yejABEF$ which was used previously. Thus, we have done the full cell wall analysis for this mutant and altered Figure 4 panels C, D and E to include this new data. We have also included as a supplementary figure (Fig S5) the chromatograms (S5A), summary table of major peptidoglycan features in this strain versus the WT (S5B) and also the full quantification of all peaks and the MS ion m/z values which were used to identify them (S5C). We believe this change improves the flow and ease of understanding the figure, since $\Delta yejB$ is not used elsewhere. We have also replaced the microscopy (panel A) to correspond.

I am missing a discussion about the peculiarity of PG recycling during apical growth. Is this possibly the reason for being vital for cell wall integrity ?

Indeed, we also wondered if this could be the reason. We have added a section to the discussion (lines 338-344):

“The requirement of YejABEF for cell wall integrity and normal growth was surprising, since *E. coli* has been shown to have no physiological defects on deletion of AmpG (7). A possible reason for this is the different ways in which these bacteria grow: while *E. coli* elongates all along the cell (8), *A. tumefaciens* elongates from one pole (1). Integrity of the newly synthesized PG is perhaps more susceptible to a lowered flux of material when synthesis is concentrated on a single area.”

Reviewer #2 (Remarks to the Author):

Bacteria of the Rhizobiales/Rhodobacterales orders of Alphaproteobacteria lack the model system muropeptide transporter AmpG, despite having other key PG recycling enzymes. The authors use a drug resistance screen to identify an alternative transporter system, YejBEF-YepA, that plays this role in the Rhizobiales phytopathogen *Agrobacterium tumefaciens*, and absence of YejBEF-YepA causes severe cell wall and growth defects. Thus, PG recycling is essential for cell wall integrity in *A. tumefaciens*. This result up-ends previous studies in model Gram negative bacteria, such as *E. coli*, where deleting genes required for PG recycling has no effect on cell growth under lab conditions. The data underscore the need to further study such pathways in important non-model systems that act in important medical and plant specific pathogens.

Most experiments are of quality and support the conclusions drawn. However, clarifications and additions to the text and figures are warranted as described below.

We thank the reviewer for their detailed read of our manuscript and their positive comments.

1. The manuscript spends a lot of words on speculations, especially in the Discussion. Instead, the results would benefit from expansion while the Discussion is reduced.

We thank the reviewer for their feedback. We have generally expanded the results section, and removed some of the discussion with a focus on the more speculative aspects.

2. The figures throughout showing ampicillin MIC test strips are not clear. The strips themselves should be removed, as they will not be readable when such figures are reduced during potential publication. The different degrees of bacterial growth on the plates are obviously different. It is not clear what the strips were meant to show relative to the growth patterns; this needs to be explained in more detail. The authors should design an overlay scale (to place on top of the images of bacterial growth on plates) that is readable and interpretable?

We understand that the intended readout of the strips (the MIC) may not be clear from the figures to some, and hence included this data for the reader in a table in Figure 3G. Indeed, the different degrees of bacterial growth are obviously different; the purpose of the strips is to convey this in a visual manner. Additionally, several mutants display MIC values which are below the readout of the strips, and so including the images allows these to be differentiated by the reader. However, to improve the direct comparison of the MIC strips, we have added a scale which indicates the major antibiotic concentrations.

3. Figure 1A PG recycling pathway needs to be expanded. Where does Fosfomycin act in the pathways shown? Where does the MurA-catalysed conversion of UDP-GlcNAc into UDP-1-carboxyvinyl-GlcNAc occur? Where is substrate for the subsequent production of UDP-MurNAc by MurB? Where does the recycling of MurNAc occur? These edits to the figure would help the reader better understand lines 101-106.

We again thank the reviewer for their helpful feedback. We have expanded the figure to include the synthesis of UDP-MurNAc from UDP-GlcNAc, an indication of where Fosfomycin acts, and the MurNAc recycling pathway from *Pseudomonas* shown with separately coloured arrows. We agree that this better allows the reader to understand our screen.

4. The diagram in Figure 2A is similarly cryptic. The legend should better justify why the red, orange, dark blue, light blue, green proteins are placed in the cytoplasm, membrane, or periplasm.

Duly noted. We have better indicated on the diagram the location of the cytoplasm, inner membrane and periplasm to help the reader interpret it, and expanded on these abbreviations in the legend. As for why these proteins were represented in their respective localisations, this comes from their annotations as either membrane-spanning proteins, periplasmic substrate binding proteins or a cytoplasmic ATP-binding protein. We have also added this information to the legend to clarify this (lines 702-704).

5. There is no legend for Fig. 2E. Fig. 2E itself seems to be missing information as only black and orange data are shown and no blue or red data are shown.

We apologise for the confusion; this is an embarrassing oversight on our part. We have added a legend for Fig. 2E (lines 709-712). The blue and red data are indeed shown, but only in the form of their black dots and not coloured bars, which are missing due to the scale. To better display the results from these strains, we have changed the colour of the black dots for all of the datapoints to match their legend colours.

6. Lines 162-163 say there is “ > 250 fold decrease in Amp MIC in the transport mutant” (Fig. 3B). How can we see this from the plates shown, especially when one cannot read the strips? Please clarify.

We agree this is not visible from the strip images, and have altered the sentence to refer to Fig. 3G instead, which includes the numerical MIC values read from the strips.

7. Lines 171-173 say that Fig. 3C will compare mutants of *yejA* and *Atu1774*, but these mutants are not found in this panel or any other panels of Fig. 3? Line 178 also says these same mutants will be compared in Fig. 3E, but they are not shown on this panel?

The authors make a conclusion about this data (lines 179-181) but where are the data?

Indeed, the comparison of Amp resistance of $\Delta yejA$ and $\Delta Atu1774$ is found in Fig. 3D, not in Fig. 3C; we have corrected this in the text. However, we are unsure what the reviewer means about Fig. 3E, since chromatograms from both of these strains are depicted alongside the $\Delta yejABEF$ mutant. We are however aware that confusion could arise from the use of the name “*yepA*” instead of *Atu1774* at this point. We have therefore attempted to clarify that these are one and the same in the Figure legend.

8. Basically, it is confusing to the reader when the authors refer to the wrong figures. Please check throughout. Additionally, lines 182 -183 refer to Fig. 3B, but presumably the authors mean to say Supp. Fig. 3B?

We apologise for the confusion. We have fixed this in the text to refer correctly to Sup. Fig. 3B (line 187) and checked the text for any other incorrect figure references; we believe that we have fixed all.

9. Lines 199-200. The authors should combine Fig. 4A and Supp. Fig. 4A to be able to compare all the data on a single graph.

We are unsure if the reviewer means to refer to Fig 4A since these are microscopy images and not related to the growth curves in Fig S4A. We assume the reviewer instead refers to Fig 4B. In this case, we agree that all of the data should be included, and thank the reviewer for their suggestion. We have updated Fig 4B to include this extra data.

10. Lines 214-235. It is not clear why the authors choose to mention particular genes of interest. Logic of choices should be justified, with cross reference to the read ratios in panel F.

The Tn-Seq screen resulted in many different candidate genes, most of which require significant thought to link to PG recycling and are thus outside of the scope of this manuscript. As such, we chose to focus on genes which could be more directly related to PG homeostasis and cell division which were beyond the cutoff value in statistical significance ($p < 0.01$) to discuss in the text, and annotated these in the figure. We have clarified this in the figure legend (lines 744-745).

11. Lines 311-315. Speculation about *A. tumefaciens* encountering antibiotics in the soil microenvironment seems unwarranted, unless all our water is contaminated?

Soil microbes such as bacteria or moulds are a major source of antibiotics including diverse beta-lactams (9), which they naturally produce and use to fight for their ecological niche (10). Indeed, *A. tumefaciens* encoded an inducible β -lactamase system long before humanity discovered the utility of penicillin. We believe speculation on this is indeed warranted, since *A. tumefaciens* encounters many other organisms in the soil environment, including β -lactam producing *Penicillium* fungi. We have clarified the putative natural source of these antibiotics in the text (line 334):

“As a soil bacterium, *A. tumefaciens* likely encounters β -lactam antibiotics in the environment produced by competing organisms to interact or fight for their niche (10), so the presence of an inducible β -lactamase system could be a way to counter these.”

12. Lines 343-345. Bacteroids are extremely large and enclosed in plant membranes. It is not appropriate to compare their morphologies with those of single mutant cells in *A. tumefaciens* that are simply swollen due to defects in PG.

We respectfully disagree that this is not an appropriate comparison, since we are comparing the relative difference between the WT and transporter mutants in each case: our *A. tumefaciens* WT and $\Delta yejABEF$ strains grown *in vitro* in normal and low-osmolarity media, and *Sinorhizobium meliloti* WT and $\Delta yejE/\Delta yejF$ strains forming bacteroids *in planta*. In each case, the transporter mutant is highly swollen and spherical compared its corresponding WT. However, we have slightly softened the language used in this section and also removed the direct comparison to our data (lines 371-373):

“Indeed, the bacteroids formed by *S. meliloti* $\Delta yejE$ and $\Delta yejF$ mutants display an aberrant, swollen morphology (11) indicating that PG recycling could be necessary to maintain cell wall integrity in *S. meliloti* during bacteroid differentiation.”

13. The rest of this paragraph to line 354 is highly speculative.

Duly noted. We have toned down the speculation at the end of this paragraph (lines 377-379).

Reviewer #3 (Remarks to the Author):

Gilmore and Cava report on the identification of an ABC transporter responsible for peptidoglycan recycling in *Agrobacterium tumefaciens*. The finding is original and unexpected because muropeptide recycling in well-studied Gram-negative model bacteria depends on an entirely different transporter, namely the proton-driven MFS transporter AmpG. Since *Agrobacterium* and related bacteria do not encode an AmpG homolog, the authors established a clever Tn-Seq screen to identify the responsible transport system in *Agrobacterium*. They show that – in contrast to *E. coli* - the transporter is vital.

The story is straightforward, the findings are novel, and the manuscript is well written. I only have a couple of comments and questions.

We thank the reviewer for their kind comments.

Questions and comments:

1. Fig. 1B: I did not understand why only a small and a single peak appears in the *nagZ* mutant whereas multiple peaks appear in the *ampD* mutant. Wouldn't one expect the opposite since NagZ acts upstream of AmpD (Fig. 1A)? Please clarify.

NagZ can indeed act upstream of AmpD, but AmpD also displays promiscuity in cleaving both anhydroMurNAc-linked and GlcNAc-anhydroMurNAc-linked muropeptides (12). We therefore likely only see the disaccharide accumulate in $\Delta nagZ$ because AmpD is still active. Similarly, NagZ can also act promiscuously on disaccharides with or without their peptide chains (13, 14). We have clarified this promiscuity in the Fig. 1A legend (lines 689-692), and thank the reviewer for pointing this out.

2. Is the *yejABEF* mutant sensitive to antibiotics other than beta-lactams or to membrane-damaging agents? Does the mutant have any other growth defects (except for the ones shown in Fig. 4)?

Deletion of the transporter was previously shown to confer a mild sensitivity to the detergent SDS in *Sinorhizobium meliloti* (11). Additionally, we tested sensitivity of our $\Delta yejB$ mutant to polymyxin B using MIC test strips but saw no difference:

We are not aware of any other growth defects.

3. Fig. 2C: Are the two gene clusters located on the same replicon (circular or linear chromosome)?

Indeed, they are both located on the circular chromosome.

4. Fig. 4A: Add length of the scale bar in the figure or legend.

We thank the reviewer for pointing this error out. We have added the length in the legend (5 μm , lines 734-735).

5. Lines 298/299: Do transcriptomics data in public databases tell us anything about the expression of these genes?

A plant infection RNA-seq showed a very mild differential regulation of *yepA* (upregulated) and *yejA* (downregulated) in *A. tumefaciens* isolated from plant tumours (15) suggesting there is some possibility of the transcriptional control of these components to facilitate PG recycling during infection.

Additionally, a later Tn-Seq experiment has shown the requirement of the transporter for fitness in colonising tomato plant roots (16), a condition which lacks transcriptomics data to our knowledge.

6. Likewise for bacteria that contain both transport system (e.g. Bradyrhizobium): Do publicly available transcriptomics data show whether both systems are (differentially) expressed under laboratory conditions or in the plant?

In general we found it difficult and time consuming to find genes in transcriptomics data due to a wide variety of different locus tags used, and poor annotation of both *ampG* and *yejBEF-yepA*. From the FitnessBrowser database of genome-wide fitness experiments (17), both *ampG* and *yejABEF* Tn mutants have similar PG-related phenotypes in *Azospirillum brasilense* under lab conditions, indicating that both are probably active under those conditions:

Group	Condition	ampG		yejB	
		AZOBR_RS09735	AZOBR_RS03820	AZOBR_RS09735	AZOBR_RS03820
stress	Bacitracin 0.25 mg/ml	-2.0	-1.7		
motility	outer cut, LB soft agar motility assay	-1.1	-1.6		
stress	Benzethonium chloride 0.003 mM	-1.3	-1.4		
nitrogen source	D-Alanine (N)	-1.3	-1.0		
stress	Vancomycin 0.03 mg/ml	-1.2	-0.8		
stress	copper (II) chloride 2 mM	-1.0	-0.8		
nitrogen source	D-Alanine (N)	-1.0	-0.8		
motility	inner cut, LB soft agar motility assay	-1.4	-0.3		
stress	Sisomicin 0.01 mg/ml	-1.0	-0.4		
carbon source	D-Gluconic Acid (C)	-0.7	-0.7		
motility	inner cut, LB soft agar motility assay	-1.3	-0.1		
stress	nitrite 10 mM	-0.8	-0.5		
stress	nitrate 100 mM	-0.5	-0.8		
nitrogen source	Glucuronamide (N)	-0.5	-0.8		
stress	Aluminum chloride 5 mM	-0.7	-0.6		
nitrogen source	L-Citrulline (N)	-0.7	-0.5		
carbon source	succinate (C)	-0.7	-0.5		
nitrogen source	Cytosine (N)	-0.5	-0.7		
nitrogen source	L-Lysine (N)	-0.6	-0.5		
lb	LB	-0.8	-0.3		
stress	D-Cycloserine 0.125 mg/ml	-0.5	-0.5		
nitrogen source	L-Citrulline (N)	-0.5	-0.5		

One RNA-seq of *A. brasilense* showed a modest upregulation of both *ampG* and *yejBEF-yepA* during “cyst” formation, a cell type formed during nutrient deprivation or dessication (18):

References

1. P. J. B. Brown, *et al.*, Polar growth in the Alphaproteobacterial order Rhizobiales. *Proceedings of the National Academy of Sciences of the United States of America* **109**, 1697–1701 (2012).
2. S. A. Woldemeskel, *et al.*, The conserved transcriptional regulator CdnL is required for metabolic homeostasis and morphogenesis in *Caulobacter*. *PLoS Genet* **16**, e1008591 (2020).
3. P. Garcia-Vello, *et al.*, Peptidoglycan from *Akkermansia muciniphila* MucT: chemical structure and immunostimulatory properties of muropeptides. *Glycobiology* **32**, 712–719 (2022).

4. Q. Cheng, J. T. Park, Substrate Specificity of the AmpG Permease Required for Recycling of Cell Wall Anhydro-Muropeptides. *Journal of Bacteriology* **184**, 6434–6436 (2002).
5. A. Aliashkevich, F. Cava, LD-transpeptidases: the great unknown among the peptidoglycan cross-linkers. *FEBS Journal* (2021) <https://doi.org/10.1111/febs.16066>.
6. L. Alvarez, B. Cordier, S. van Teeffelen, F. Cava, Analysis of Gram-negative Bacteria Peptidoglycan by Ultra-performance Liquid Chromatography. *Bio-protocol* **10**, e3780–e3780 (2020).
7. J. T. Park, T. Uehara, How Bacteria Consume Their Own Exoskeletons (Turnover and Recycling of Cell Wall Peptidoglycan). *Microbiology and Molecular Biology Reviews* **72**, 211–227 (2008).
8. E. Kuru, *et al.*, In Situ Probing of Newly Synthesized Peptidoglycan in Live Bacteria with Fluorescent D-Amino Acids. *Angewandte Chemie International Edition* **51**, 12519–12523 (2012).
9. R. E. de Lima Procópio, I. R. da Silva, M. K. Martins, J. L. de Azevedo, J. M. de Araújo, Antibiotics produced by *Streptomyces*. *The Brazilian Journal of Infectious Diseases* **16**, 466–471 (2012).
10. M. I. Abrudan, *et al.*, Socially mediated induction and suppression of antibiosis during bacterial coexistence. *Proc Natl Acad Sci U S A* **112**, 11054–11059 (2015).
11. Q. Nicoud, *et al.*, Sinorhizobium meliloti Functions Required for Resistance to Antimicrobial NCR Peptides and Bacteroid Differentiation. *mBio* **12**, e00895-21 (2021).
12. C. Jacobs, *et al.*, AmpD, essential for both β -lactamase regulation and cell wall recycling, is a novel cytosolic N-acetylmuramyl-L-alanine amidase. *Molecular Microbiology* **15**, 553–559 (1995).
13. W. Vötsch, M. F. Templin, Characterization of a β -N-acetylglucosaminidase of *Escherichia coli* and Elucidation of Its Role in Muropeptide Recycling and β -Lactamase Induction*. *Journal of Biological Chemistry* **275**, 39032–39038 (2000).
14. Q. Cheng, H. Li, K. Merdek, J. T. Park, Molecular Characterization of the β -N-Acetylglucosaminidase of *Escherichia coli* and Its Role in Cell Wall Recycling. *Journal of Bacteriology* **182**, 4836–4840 (2000).
15. A. González-Mula, *et al.*, Lifestyle of the biotroph *Agrobacterium tumefaciens* in the ecological niche constructed on its host plant. *New Phytol* **219**, 350–362 (2018).
16. M. Torres, *et al.*, *Agrobacterium tumefaciens* fitness genes involved in the colonization of plant tumors and roots. *New Phytologist* **233**, 905–918 (2022).
17. M. N. Price, *et al.*, Mutant phenotypes for thousands of bacterial genes of unknown function. *Nature* **557**, 503–509 (2018).
18. E. A. Malinich, C. E. Y. 2018 Bauer, Transcriptome analysis of *Azospirillum brasilense* vegetative and cyst states reveals large-scale alterations in metabolic and replicative gene expression. *Microbial Genomics* **4**, e000200.

Reviewer #1 (Remarks to the Author):

The revised version of the ms by Gilmore and Cava is significantly improved. I am satisfied with the changes the authors made.

I have only one remark that should be considered in the final version: the abbreviations are in some case insufficiently defined:

e.g. UDP-M5 occurs in Fig. 2E, but is defined in the text much later, and anhMurNAc-P4 is not clearly defined. Why 5 and P4 ? Please check ! It may be helpful to include these abbreviations in Table S3 and refer to them in the text.

Congratulations to the authors for this nice piece of work.

Reviewer #2 (Remarks to the Author):

The authors have addressed my comments satisfactorily. However, I wish to make further comments regarding Figure 1A. The lower part of the diagram cannot be understood or comprehended unless one enlarges the figure to a significant extent. The lower bottom right gives a key for the diagrams for 10 different compounds shown in the pathways on the left. But the diagrams of the bottom 5 compounds are extremely small (basically tiny dots of different colors), both in the "key" and in the pathway(s) to the left. A possible solution would be to make Figure 1 A into a supplementary figure where all muro-peptides could be drawn at a larger scale.

Reviewer #3 (Remarks to the Author):

I am satisfied with the revisions.

Reviewer #1 (Remarks to the Author):

The revised version of the ms by Gilmore and Cava is significantly improved. I am satisfied with the changes the authors made.

I have only one remark that should be considered in the final version: the abbreviations are in some case insufficiently defined:

e.g. UDP-M5 occurs in Fig. 2E, but is defined in the text much later, and anhMurNAc-P4 is not clearly defined. Why 5 and P4 ? Please check ! It may be helpful to include these abbreviations in Table S3 and refer to them in the text.

We thank the reviewer for the feedback and ideas. We have now defined UDP-M5 in the introduction, and added it to Table S3. We have added abbreviations to Table S3 for all of the compounds represented there.

Congratulations to the authors for this nice piece of work.

Reviewer #2 (Remarks to the Author):

The authors have addressed my comments satisfactorily. However, I wish to make further comments regarding Figure 1A. The lower part of the diagram cannot be understood or comprehended unless one enlarges the figure to a significant extent. The lower bottom right gives a key for the diagrams for 10 different compounds shown in the pathways on the left. But the diagrams of the bottom 5 compounds are extremely small (basically tiny dots of different colors), both in the "key" and in the pathway(s) to the left. A possible solution would be to make Figure 1 A into a supplementary figure where all muro-peptides could be drawn at a larger scale.

We appreciate the reviewer's feedback on the comprehensibility of the schematic diagram. We prefer to keep this diagram in the main text, due to its importance to the reader in understanding both the screen and the process of PG recycling. However, we have increased the size as far as possible to reduce the need to enlarge it.

Reviewer #3 (Remarks to the Author):

I am satisfied with the revisions.